# Sensory Processing Disorders in Children and Adolescents: Taking Stock of Assessment and Novel Therapeutic Tools

**DOI:** 10.3390/brainsci12111478

**Published:** 2022-10-31

**Authors:** Noemi Passarello, Vincenza Tarantino, Andrea Chirico, Deny Menghini, Floriana Costanzo, Pierpaolo Sorrentino, Elisa Fucà, Onofrio Gigliotta, Fabio Alivernini, Massimiliano Oliveri, Fabio Lucidi, Stefano Vicari, Laura Mandolesi, Patrizia Turriziani

**Affiliations:** 1Department of Humanities, “Federico II” University, Via Porta di Massa 1, 80138 Naples, Italy; 2Department of Psychology, Educational Sciences and Human Movement, University of Palermo, Via delle Scienze, Ed. 15, 90128 Palermo, Italy; 3Department of Social and Developmental Psychology, Faculty of Medicine and Psychology, “Sapienza” University of Rome, Via dei Marsi 78, 00185 Rome, Italy; 4Child and Adolescent Neuropsychiatry Unit, Department of Neuroscience, Bambino Gesù Children’s Hospital, IRCCS, Viale di San Paolo, 15, 00146 Rome, Italy; 5Institut de Neurosciences des Systèmes, Aix-Marseille University, 27 Bd Jean Moulin, 13005 Marseille, France; 6Department of Life Sciences and Public Health, Catholic University, Largo Francesco Vito 1, 00168 Rome, Italy

**Keywords:** perception, cognition, self-report questionnaires, Dunn’s framework, neurodevelopment

## Abstract

Sensory processing disorders (SPDs) can be described as difficulty detecting, modulating, interpreting, and/or responding to sensory experiences. Because SPDs occur in many individuals with autism spectrum disorder and in other populations with neurodevelopmental disorders, it is important to distinguish between typical and atypical functioning in sensory processes and to identify early phenotypic markers for developing SPDs. This review considers different methods for diagnosing SPDs to outline a multidisciplinary approach useful for developing valid diagnostic measures. In particular, the advantages and limitations of the most commonly used tools in assessment of SPDs, such as caregiver reports, clinical observation, and psychophysical and neuroimaging studies, will be reviewed. Innovative treatment methods such as neuromodulation techniques and virtual reality will also be suggested.

## 1. Introduction

In recent decades, there has been an increased interest in the role that sensory processing plays in development [1]. Sensory processing allows us to organize information from the body and the environment and influences the way we interact with our physical and social surroundings. From birth, sensory processing influences infants’ actions through reflexive motor actions and state regulation [2]. The concept of sensory integration proposed by Ayres [3] can be viewed as the conceptual equivalent of sensory processing. Sensory integration relates to multimodal processing, which supports the formation and retrieval of multisensory perceptions within the central nervous system, to enable sensory information processing and organization. Some children show clinically significant difficulties regulating their response to sensation in a way that interferes with daily life activities and routines as well as learning [4]. It has been observed that 5%–13% of children from 4 to 6 years old are affected by these sensory disorders and that they suffer from debilitating social and emotional consequences due to their sensory impairments [5]. In this line, sensory processing disorders (SPDs), involving difficulty detecting, modulating, interpreting and/or responding to sensory experiences, have been increasingly studied [6]. Several sensory processing systems are impaired in SPDs, including auditory, visual, vestibular, touch, multisensory, taste, and smell. SPDs manifest as extreme reactions to sensory stimuli, in a range of “fight, flight or freeze” behaviours such as aggression, withdrawal, or preoccupation with the expectation of sensory input [7]. These behavioural responses to stimuli can be categorized in three different patterns: sensory over-responsiveness, sensory under-responsiveness, or craving for sensory input. Sensory over-responsive children exhibit exaggerated response to stimuli that typical-development children (TD) would find tolerable; they show tactile defensiveness and gravitational insecurity; they may react negatively to certain food groups, and they avoid social situations. Meanwhile, under-responsive children lack the ability to respond to sensory cues, show low arousal, seem uninterested, and daydream, and they often have low endurance and may mouth objects. Finally, children with sensory craving (or sensory seeking) constantly touch, move, crash into objects, and have little to no awareness of personal space or boundaries. These children tend to be clumsy and awkward and may also demonstrate decreased safety awareness.

There is evidence that sensory symptoms affect attention and communication abilities [8,9,10]. Sensory symptoms also negatively impact family life [11,12] and cause increased parental stress [13]. SPDs are also associated with high risk of internalizing and externalizing problems as well as health issues over the course of lifetime. For instance, sensory over-responsivity is associated with chronic gastrointestinal symptoms [14] and sleep disturbance [15,16]. Children with SPDs may also experience food fussiness and eating issues [17,18].

Recent studies indicates that SPDs are linked to several neurodevelopmental disorders, especially autism spectrum disorder (ASD) and attention deficit hyperactivity disorder (ADHD) [19,20,21]. Atypical sensory experiences are believed to occur in up to 90% of individuals with ASD [22] and in 50-64% of children with ADHD [23,24]. Therefore, SPDs could potentially serve as early diagnostic markers of these neuropsychiatric disorders. Children can also experience sensory processing dysfunction outside the clinical conditions of ASD or ADHD. Literature refers to this as isolated SPDs [25].

SPDs, however, remain poorly understood. A possible reason for this is the lack of precise criteria and diagnostic tools useful to identify SPDs specific sensory/behavioural pattern. Further, as we will discuss in this review, another possible account of the lack of knowledge of SPDs concerns researchers’ focus on mainly reporting sensory domain deficits. Evidence is emerging that even other domains, such as perception and cognition, could be involved in stimulus processing [25]. As a result of the scientific and clinical approach taken so far, SPDs have been diagnosed and treated through methods that only consider sensory issues, while the cognitive components are frequently neglected. This could be the reason why traditional assessment methods, such as caregiver- and teacher-report questionnaires, fail to provide a comprehensive overview of SPDs symptoms and lead to contradictory findings. Moreover, scientific literature does not emphasize enough the importance of different techniques integration as a method to obtain more accurate representation of the SPDs. Experimental studies rarely combine different techniques or analyse the advantages and disadvantages of each of them to develop diagnostics and treatment protocols that are more sensitive to detection of atypical sensory experience.

The aim of this narrative review will provide a detailed analysis of methods and tools used for SPDs assessment considering behavioural and neuroimaging studies, and we discuss each approach’s strengths and weaknesses. As we intend to focus on methodologies, we do not discuss studies’ findings. We also propose the use of innovative techniques, such as neuromodulation and virtual reality, that could be crucial support techniques for the study and the treatment of sensory processing in typical and atypical development. To our knowledge, this is the first review that discusses diagnostic and research tools and methods for assessing sensory abnormalities in children and adolescents. By highlighting the strong need for integrated scientific methods and the use of technological innovations to study SPDs, this paper makes a valuable contribution.

## 2. Method

A total of 56 studies were selected to discuss SPDs assessment and therapy methods. We carried out our search using PubMed/Medline, Scopus, and Web of Science databases. Search terms included a combination of “sensory processing disorders”; “neurodevelopment”; “atypical development”; “self-report questionnaires”; “neuromodulation”, “virtual reality”; and “neuroimaging”. Each of these studies employed different research methods to examine atypical sensory experiences in children and adolescents with SPD or other neurodevelopmental disorder (ASD, ADHD) characterized by sensory abnormalities. Only articles in indexed peer-reviewed journals and written in English were included.

This narrative review was conducted to illustrate the strengths and weaknesses of measures used in SPD studies and evaluations. Considering the vast amount of literature, the purpose of this review is not to analyse the methodologies used systematically but to highlight their advantages, suggest alternatives to compensate for their limitations, and stress how combined methods will help us better understand the multidimensional nature of SPD.

## 3. Caregiver and Self-Report Questionnaires

Most studies on SPDs are based on caregiver or self-report measures, which are also the useful approach for diagnosing of these disorders. Over the years, a variety of questionnaires have been designed to examine this phenomenon. In this section, after having described in detail the most used ones, their limits and potentials will also be discussed. In Table 1, caregiver and self-report questionnaires are reported.

### 3.1. Sensory Profile (SP)

The most widely used questionnaire is the Sensory Profile (SP) [26], a caregiver-report questionnaire that is designed to function as a part of a broader assessment of a child’s functioning, which may include other observations. This questionnaire can be administered to children of 3–16 years old. It contains 125 items organized into three sections: sensory processing, sensory modulation, and behaviour and emotion responses. The sensory processing section of the questionnaire measures the child’s response to six different sensory processes: auditory, visual, vestibular, tactile, multisensory, and oral. The sensory modulation section provides information regarding the child’s regulation of sensory input through facilitation or inhibition of different types of responses. Five different areas of sensory modulation can be identified: endurance/tone, position/movement, movement effecting activity level, and modulation of sensory input. The behavioural and emotional responses section provides information regarding the child’s behaviours that occur as an outcome of sensory processing. The three areas include emotional/social responses, behavioural outcomes of sensory processing, and items indicating threshold for response. The items on the questionnaire can also be grouped into the following nine different factors that describe children by their response to sensory input (over- or under-responsive): sensory seeking, emotionally reactive, low endurance/tone, oral sensory sensitivity, inattention/distractibility, poor registration, sensory sensitivity, sedentary, and fine/motor perceptual. Results can also be grouped into four quadrant scores, including registration, seeking, sensitivity, and avoiding [26]. Caregivers report the frequency of the behaviours in question using a 5-point Likert scale ranging from “1 = always” to “5 = never”, with lower scores indicating higher sensory sensitivity.

### 3.2. Child Sensory Profile 2 (SP-2)

The Child Sensory Profile 2 (SP-2) [27] is an 86-item caregiver-report measure of a child’s sensory processing characteristics. The SP-2 preserves most of the structure of the original SP; it measures the four quadrants of sensory processing patterns: sensation seeking, sensory avoiding, sensory sensitivity, and sensory registration. Additionally, the questionnaire assesses six sensory systems (i.e., auditory, visual, touch, movement, body position, and oral) and three behavioural sections (i.e., conduct, attention, and social/emotional). Differing from the original, the Likert scoring of the SP-2 represents “1 = almost never” to “5 = almost always” and also includes a “0 = does not apply” response category.

### 3.3. Short Sensory Profile (SSP)

The SP is also available in a short form, the Short Sensory Profile (SSP) [28], that is a 38-item designed to assess children’s responses to sensory stimuli. The questionnaire is composed by three subscales that assess children’s tactile sensitivity (e.g., avoids going barefoot, especially in grass and sand), taste/smell sensitivity (e.g., avoids tastes or food smells that are typically part of a child’s diet), and visual/auditory sensitivity (e.g., covers eyes or squints to protect eyes from light). Caregivers responded to items on a 5-point Likert scale ranging from “1 = always” to “5 = never”, with lower scores indicating higher sensory sensitivity. SSP total scores can range from a minimum of 38 (the greatest frequency of sensory symptoms) to 190 (no sensory symptoms).

### 3.4. Sensory Experiences Questionnaire (SEQ)

Like the previous tools described, the Sensory Experiences Questionnaire (SEQ version 3.0) [29] is a brief caregiver-report instrument designed to evaluate behavioural responses to common everyday sensory experiences in children ages 5 months through 6 years. The SEQ 3.0 has 105 items that measure the frequency of sensory behaviours across sensory response patterns (i.e., hypo-responsiveness, hyper-responsiveness, sensory seeking, and enhanced perception), modalities (i.e., auditory, visual, tactile, gustatory, and vestibular), and social or non-social contexts. Caregivers assess children behaviour in a 5-point Likert scale. Its primary purpose is to characterize sensory features in young children with autism and to discriminate patterns of hypo- and hyper-responsiveness among those with autism, developmental delay, or typical development. The SEQ is intended to be used either for research purposes or as a clinical supplement to traditional developmental or diagnostic assessments that typically do not tap sensory features.

### 3.5. Sensory Rating Scale (SRS)

The Sensory Rating Scale (SRS) [30] is another caregiver report measure that is used to identify and quantify sensory responsiveness in children 0 to 3 years of age. The final version of the SRS has two forms used to assess sensory processing in two different age brackets. Form A consists of 88 questions and is appropriate for use from birth to 8 months. Form B consists of 136 questions and is targeted at an older age range, 9 months to 3 years. Each item on the SRS is scored on a 5-point Likert scale, with scores of 4 and 5 considered as high risk for each of the sensory defensive behaviours.

### 3.6. SensOR Inventory (SensOR)

There is a specific scale for sensory over-responsiveness (SOR), the SensOR Inventory (SensOR) [31], a self/caregiver-report tool that identifies individuals with SOR in different sensory domains. It can be applied to a wide population (3–55 years), and it is composed of 76 items divided into eight sensory categories: tactile—textures, tactile—activities of daily living, auditory settings, auditory-specific, visual, olfactory, movement—proprioceptive, and food—textures/eating. The SensOR is currently the only available scale that clearly differentiates SOR from typical behaviour. Each item is scored as a “1” if behaviours or activities “bother” the individual or a “0” if the behaviours or activities are not bothersome.

### 3.7. Sensory Processing Measure (SPM)

It is possible to use the Sensory Processing Measure (SPM) [32] to assess sensory behaviours at school. This is a caregiver-/teacher-report questionnaire in two versions: one for home (SPM-home) and one for school (SPM-school). The version for children aged from 3 to 10 years is composed by 62 items and assesses different domains: social participation, vision, hearing, touch, body awareness, balance and motion, and motor planning. SPM is a tool for the assessment of sensory processing, praxis, and social participation in different school environments. The purpose of this tool is to provide teachers with information regarding sensory facilitators and barriers in order to help students perform better.

### 3.8. Infant/Toddler Sensory Profile (ITSP)

Assessment of sensory processing abilities in very young children can be conducted through the Infant/Toddler Sensory Profile (ITSP) [33], a caregiver-report questionnaire used with infants and toddlers from birth to 36 months of age. The ITSP uses the primary caregiver’s knowledge of their child’s functional performance in the realistic setting beyond the assessment room walls. The primary features of this model include the consideration of a person’s neurological threshold (i.e., reactivity/responsivity), consideration of responding or self-regulation strategies (response), and consideration of the interaction among thresholds and responding strategies. Neurological threshold can be defined as the number of stimuli required for neural systems to respond. Items on the ITSP questionnaire describe age-appropriate behaviours or responses to various sensory experiences within the different sensory systems. Items are grouped into six sensory sections: general processing, auditory processing, visual processing, tactile processing, vestibular processing, and oral sensory processing. As the infant responds to the items featured in each section, different patterns of sensory responsiveness will be displayed across the four quadrants of the model.

### 3.9. Diagnostic Interview for Social and Communication Disorders (DISCO)

One clinical interview measure that assesses sensory symptoms in detail is the Diagnostic Interview for Social and Communication Disorders (DISCO) [20], a clinical interviewer-based schedule designed for use with parents and carers. Its purpose is to elicit information relevant to the broad autistic spectrum to assist clinicians in their judgement of an individual’s level of development and specific needs. Diagnostic categories can also be derived from the DISCO information by using a set of algorithms. In addition to many other items relating to developmental skills and atypical behaviour, the DISCO records in detail the patterns of sensory features seen in children of any age and with any degree of impairment. The DISCO uses 21 items related to sensory abnormality that are separated into three groups: proximal (e.g., touch, taste, smell, and kinaesthetic) (14 items), auditory (3 items), and visual (4 items). Additionally, there are items relating to kinaesthetic, oral, and touch atypicality throughout the DISCO. The type and degree of every sensory abnormality is explicitly recorded within the DISCO diagnostic profile, making it possible to examine the extent to which abnormalities occur in different sensory domains. The sensory items used in the DISCO also differ from the items of the Sensory Profile. DISCO items have been chosen based on many years of clinical observation as items that are frequently observed in people with social and communication disorders. Although there is overlap in the content of sensory items, the way that information is collected by the Sensory Profile questionnaire compared with the DISCO interview is very different in terms of not only the methods of collecting data but also the measurement scales used for analysis [20].

### 3.10. Adolescent/Adult Sensory Profile (AASP)

Adolescent/Adult Sensory Profile (AASP) [34] is another widely used measurement to assess SPD in adolescent and young adults (from 11 to up 65 years). Its use highlights how much SPDs are present even outside childhood. AASP is a self-report questionnaire that maintain Dunn’s structure in four sensory quadrants (sensation seeking, low registration, sensation avoiding, and sensory sensitivity) [26]. It consists of a 60-item self-report questionnaire assessing levels of sensory processing in everyday life. Although AASP is a useful clinical tool, it does not permit the investigation of each sensory domain separately.

The use of caregiver-/self-report questionnaires has always been fundamental in child development research. The primary benefit of relying on caregiver reporting is that it works to overcome limitations caused by a lack of children’s verbal abilities and metacognitive skills. Additionally, since children’s behaviour is highly sensitive to contextual, relational, and sociocultural influences, continuous and long-term observation by caretakers might be more reliable than a single observation by a clinician. Lastly, psychometric properties of all the described assessment tools have been rigorously evaluated. Therefore, these tools are highly reliable and easy to use. However, they are limited in gathering clinical and experimental insights about SPDs heterogeneous patterns. It is important to note that these questionnaires emphasize the experiential component of sensory perception and leave out cognitive processing, which is equally deficient in SPDs. Furthermore, most of them do not consider other cognitive functions, such as attention, that are known to be involved in modulating and processing sensory input [35]. A further limitation is the lack of detailed descriptions of the specific difficulties of each sensory domain, with little to no focus on multisensory integration. Though parents’ reporting of their child’s behaviour can be reliable, responses may also be subject to bias [36]. Consequently, a diagnosis based solely on a parent/caregiver’s report may be incomplete. In most of the questionnaires described above, environment is rarely considered as an influence on sensory experience. Thus, these tools lack ecological validity. Lastly, caregiver-report questionnaires do not account for the alterations in the translation of neural firing into sensory symptoms, nor do they acknowledge the role of perceptual, cognitive, and emotional factors [37].

## 4. Behavioural and Psychophysiological Approach

The behavioural approach can be used to study SPDs by observing a child’s behaviour when interacting with different stimuli, such as during clinical observation. On the other hand, a psychophysiological approach uses threshold testing to study the responses of children with SPDs to sensory stimuli. Furthermore, electroencephalography (EEG) in conjunction with behavioural paradigms can provide information on the brain activity correlated to SPD. In this paragraph, we will discuss these approaches.

### 4.1. Clinical Observation

SPDs can be better examined through clinical observation of children’s behaviour. The most commonly used clinical observational tool is the Sensory Integration and Praxis Test [38]. The Sensory Integration and Praxis Test is standardized on children ages and assesses the sensory integration processes underpinning learning and behaviour. It is composed by 17 brief tests that analyse how children organize and respond to sensory inputs, and it reveals problems associated with learning disabilities and emotional disorders.

Another widely used tool is the Sensory Processing Assessment for Young Children (SPA) [39], a 20-min play-based behavioural observation assessment performed in a semi-structured format. The SPA, designed for children between 9 months and 6 years of age, involves observation of the child’s reaction to tactile, auditory, and visual stimuli while interacting with toys and unexpected sensory stimuli. By examining how the child approaches or avoids novel sensory toys, determining how they react to social and non-social stimuli in the environment and measuring their accuracy, the SPA is designed to yield useful information about how the child learns. The SPA also examines self-stimulatory behaviours, including covering ears with hands or arms, hand flapping, finger mannerisms, mouthing non-food objects, smelling non-food objects, other repetitive motor movements, and repetitive object manipulation. It contains three subscales: avoidance, orienting, and defensive. Each scale has a sub-score for reactions to social and non-social stimuli. The orienting subscale relates to hypo-responsiveness to stimuli, while the avoidance subscale relates to hyper-responsiveness.

Although clinical observation overcomes biases related to the subjectivity of caregiver reports, it still has several limitations. Primarily, most behavioural observation paradigms present a limited range of sensory stimuli, often excluding multisensory and social stimuli. Cognitive processes involved in sensory processing are always neglected as well as neurophysiological correlates. Finally, these tools still lack ecological validity since the experimental/clinical setting may might prevent children from behaving as they would in everyday life.

### 4.2. Threshold Testing and Dunn’s Framework

SPDs can be studied by examining the neural thresholds and behavioural patterns that accompany sensory processing [3,26]. Dunn’s Sensory Processing Framework theorizes sensory processing as two continua of responses to environmental stimuli [33]. According to this view, people respond differently to sensory information based on how soon they detect (threshold) and how they manage (self-regulation) sensory stimuli. Threshold ranges from high (e.g., slow to detect) to low (quick to detect), and self-regulation ranges from passive (not bothered by stimuli) to active (reactive to stimuli) [24]. These two continua interact to describe four sensory processing patterns: poor registration, sensitivity to stimuli, sensation seeking, and sensation avoiding. Children with poor registration have difficulty perceiving stimuli due to high thresholds, so they act accordingly and tend to appear dull and uninterested; those characterized by sensitivity to stimuli have low thresholds and tend to be hyperactive or distractible; sensation seekers have high thresholds and may engage in behaviours to increase even more their sensory experiences; while sensation avoiders have low thresholds and avoid behaviours that can arouse them, appearing resistant and unwilling to participate. It is precisely from Dunn’s Sensory Processing Framework that various stimulus–response paradigms have been developed to focus the smallest amount of signal (such as the smallest difference in temperature) that is reliably perceptible by the individual [24].

Some evidence has shown that threshold testing can be used to assess SPDs [40]. Children with SPD show a higher static detection threshold, a mechanism that depends on GABAergic feed-forward and is crucial for the modulation of cortical activity [41]. This suggests that SPD children’s behaviour might be caused by abnormal feed-forward inhibitory mechanisms [42,43]. Research into the link between GABA and sensory function in SPDs is increasing although the literature is still developing. Combining threshold testing with neurophysiological instruments such as electroencephalography (EEG) and event-related potentials (ERPs) could shed some light on features of SPDs.

It is important to note that threshold testing methods are mainly used for research purposes in order to understand physiological mechanisms underlying the disorder. Although they may have diagnostic purposes, the clinical observation and the questionnaires remain the most used methods for assessment of SPDs. However, it is precisely due to the results of psychophysiological methods that we can now envision an integrated approach to the study and assessment of sensory atypical experiences.

### 4.3. Electrophysiological Measures

EEG is a method to record electrical activity on the scalp that has been shown to represent underneath brain activity. Event-related potentials (ERPs) represent small changes in the scalp-recorded electroencephalogram caused by sensory stimulation or a motor act. These electrophysiological measures have been used in several studies of SPDs to examine the brain’s response to different types of sensory stimulation, providing mixed results [44]. Our goal here is to report the studies that have produced the most significant results, illustrating how electrophysiological measures can serve as a valuable tool for studying SPDs. Donkers et al. [45], using the presentation of sequences of repetitive stimuli infrequently interrupted by a deviant stimulus during EEG in ASD children and TD children of ages 4 to 12, found marginally attenuated amplitude of P1 and N2 components to standard tones and attenuated amplitude of P3a component to novel sounds in ASD versus TD, thus suggesting greater sensory seeking behaviours in children with ASD. Based on these findings, it appears that neural responses to sensory information as well as attention-orientation information are attenuated through complex mechanisms that contribute to selective sensorimotor behaviours in autism. Even in the visual domain, the event-related potentials have evidenced abnormally large responses to task irrelevant stimuli, particularly in parieto-occipital and frontal regions in children with ASD in comparison to TD [46], suggesting that ASD difficulty in discriminating stimuli and motor response errors seems to be caused by poor inhibitory cortex input to the visual processing areas.

Miyazaki et al. [47] used a multimodal evoked potential (S-SEP) system to study tactile processing during sleep in SPD children. When compared with TD children, SPD individuals showed delayed peak latency for N20 component; prolonged inter-peak latency between P13/14 and N20 components; and abnormally large SEP amplitude elicited by both left and right stimulations. This same technique was used to record evoked potentials elicited by visual stimulation (VEPs) [48]. Children with and without ASD were shown a series of visual stimuli. The contrast-sweep conditions (bright or dark isolated-checks) were used to elicit VEPs. ASD displayed deficits in low-contrast responses at the stimulus frequency of 12.5 Hz. In addition, they displayed significantly higher levels of neural noise-to-signal ratio than controls. For the response at the stimulus frequency, the group with ASD produced a relatively constant level of noise across the tested ranges of contrast, with higher levels of noise than controls at low contrasts and approximately equal levels of noise at moderate-to-high contrasts. These processing dysfunctions have also been evidenced in older studies. For example, Kemner et al. [49] by means of ERPs found larger visual P2/N2 and larger visual and somatosensory P3 to novel stimuli in children with SPD, evidenced thus a limited ability to process novel information.

Electrophysiological methods also allow for the analysis of multisensory integration, which is essential in these disorders. To test whether the failure to integrate across the senses is associated with atypical sensory processing, Molholm et al. [50] investigated whether children with SPD integrate multisensory input differently than TD with similar ages or than children with ASD. Participants performed a simple reaction-time task to the occurrence of auditory, visual, and audiovisual stimuli presented in random order while high-density electrical brain activity was registered. The group with SPD showed lack of multisensory integration effect compared to both ASD and TD. This result proved the existence of an impairment in communication across the sensory systems, which fits well with the SPD phenotype of maladaptive responses to the sensory environment.

In conclusion, the ERP technique emerges as a sensitive tool for detecting group differences in sensory processing, involving both early stages/basic processes as well as later/more cognitive processes (such as inhibition of irrelevant stimuli) and multisensory integration. Unfortunately, EEG’s poor spatial resolution makes it impossible to pinpoint the precise source of activity or to distinguish between activities originating from different but closely adjacent sites. Furthermore, it is not always easy to use this method with children since it is not always possible to use wireless devices that are less sensitive to the movements a child might make in a state of agitation.

## 5. Neuroimaging Approach

Neuroimaging techniques, both structural and functional, have made a significant contribution in explaining the complexity of sensory processing. They provided us with insights into the integrated process of sensory perception and enabled the identification of multiple neural networks involved in SPDs. In this section, we will examine the neuroimaging techniques that have contributed most to the study of SPD, among which diffusion tensor imaging (DTI), magnetic resonance spectroscopy (MRS), functional magnetic resonance (fMRI), and magnetoencephalography (MEG) have played a fundamental role.

### 5.1. Diffusion Tensor Imaging (DTI) and Magnetic Resonance Spectroscopy (MRS) Measures

Diffusion tensor imaging (DTI) is a relatively new imaging technique that can be used to examine white matter in the brain. Using DTI, the orientation and direction of white matter fibre tracts can be visualized and quantified [51]. This technique has been used to study the neural correlates of SPD. Owen et al. [52] analysed the white matter of adolescents with SPD and TD through a whole-brain, data-driven approach. They found a decreased fractional anisotropy (FA) and an increased mean diffusivity (MD) and radial diffusivity (RD), reflecting reduced microstructural integrity, in the posterior white matter. Structural differences underlying auditory and tactile over-responsivity were found by Tavassoli et al. [22]. In particular, the authors evidenced a decreased FA in children with SPDs in the posterior body and isthmus of the corpus callosum, the left posterior thalamic radiations, and the posterior portion of the left superior longitudinal fasciculus. Pryweller et al. [51], focusing on white matter fibre tracts involved in sensory processing, found that FA was decreased in children with ASD and with SPDs. Finally, Payabvash et al. [53] studied the correlation between white matter microstructure and connectivity and SPDs by applying machine-learning algorithms for identification of children with SPDs based on DTI/tractography metrics. These approaches show that applying machine learning algorithms to these connectivity metrics could represent a devise for novel imaging biomarkers for neurodevelopmental disorders. DTI represents a promising tool to study and visualize white matter in SPDs children. However, it suffers from inherent artifacts and limitations. A major drawback is the partial volume effect as well as the model’s inability to handle non-Gaussian diffusion. Nevertheless, when used in conjunction with functional brain mapping, DTI is an efficient tool for obtaining comprehensive, non-invasive, functional anatomy maps of the brain.

Recent advances in magnetic resonance spectroscopy (MRS) have given important insights into SPDs. MRS is a technique that allows non-invasive quantification of a variety of neurochemicals within a localized region of tissue. Single-voxel MRS (where neurochemicals are acquired from one region of tissue only) can be used to investigate changes in gamma-aminobutyric acid (GABA) and glutamate (Glu), the major cortical neurotransmitters, in response to transcranial stimulation [43]. To this day, MRS is the only non-invasive methodology that can measure GABA concentrations in cortical regions. Studies have found that GABA transmission is crucial in SPD. GABAergic inhibition plays a significant role in defining the selectivity of cortical responses to behaviourally relevant stimuli. Puts et al. [54,55], by comparing GABA measurements in the sensorimotor areas with similar measurements in an occipital region, found a significant correlation between GABA concentration and the tactile frequency discrimination threshold only in the sensorimotor cortex voxel in individuals with SPD. Harada et al. [56] estimated GABA spectra from two separate voxels placed at basal ganglia and frontal lobe locations and observed the ratio of both GABA/N-acetyl aspartate and GABA/Glu were significantly lower in frontal lobe voxels only in a population of children with ASD. Gaetz et al. [40] found that GABA concentration was significantly reduced in motor and auditory areas in children with ASD but not significantly differed in visual areas with TD. This innovative evidence proves that MRS is a useful and widely available technique for the non-invasive measurement of tissue metabolism. However, there are some limitations in this tool, notably its spectral resolution, which is defined as the number and dimension of specific wavelength intervals of electromagnetic radiation it can measure [57].

### 5.2. Functional Magnetic Resonance (fMRI) Measures

Significant contributions to the study of SPDs have been provided by functional neuroimaging studies. A common paradigm used in fMRI to test children with SPDs is the Embedded Figures Test (EFT). During this task, individuals are asked to locate a simple figure that is embedded in a more complex configuration. To do this, participants must mentally decompose the complex figure into its structural components and decide whether some components match the target figure [58]. In a study performed by Lee et al. [58], children with ASD showed activation in only a subset of the cortical network observed in children with TD, while prefrontal cortex as well as the ventral temporal cortex remained inactive, supporting the model of weak top-down transmission. Moreover, Damrala et al. [59] found frontal-posterior functional under connectivity in children with ASD during the EFT. ASD showed more activation in brain areas typically involved in visuospatial processing (bilateral superior parietal and right occipital), while the control participants exhibited more left-lateralized activation in executive and working memory regions (dorsolateral prefrontal cortex).

Ring et al. [60] found that children with ASD showed diminished blood-oxygenation-level-dependent (BOLD) responses in the posterior cingulate cortex and the insula, particularly for pleasant and neutral tactile stimuli, compared to TD. Similarly, adolescents with over-responsiveness to sensory stimuli (SOR) showed impaired functional connectivity and difficulties in multisensory integration [61,62,63]. In these studies, adolescents with SOR and control adolescents were exposed to three stimulus conditions in a counterbalanced block design paradigm. These stimuli included an auditory condition, a tactile condition, and a joint condition, in which the auditory and tactile stimuli were presented simultaneously. Adolescents with SOR displayed stronger activation in primary sensory cortices and amygdala. This activity was positively correlated with SOR symptoms after controlling for anxiety. Cummings et al. [64] studied gender differences in adolescents with ASD and SOR using fMRI and examined the relationship between SOR and resting-state functional connectivity in brain areas involved in directing attention. Parent- rated SOR symptoms were not different for males and females, but the SOR and salience network connectivity was significantly different between sexes. Relative to females, males with ASD showed stronger association between SOR and increased connectivity between the salience and primary sensory networks, suggesting increased allocation to sensory information. On the other hand, in females with ASD, SOR was more strongly related to increased connectivity between SN and the prefrontal cortex. According to these findings, the underlying mechanism of SOR in ASD is sex-specific, explaining the difference in diagnosis rates and symptom profiles between males and females with ASD. Another study by Millin et al. [65] used an fMRI protocol to measure responses children with ASD to repeated audiovisual stimulation in early visual and auditory cortical areas. They recorded brain activity in response to simultaneous audio (white noise) and visual (checkerboard) stimulation. Children with ASD and TD were instructed to use the index finger of their dominant hand to press a button, as quickly as possible, following the appearance of each stimulus. Both the visual and auditory cortices produced the same initial transient responses. However, in auditory but not in visual cortex, the post-transient sustained response was greater in individuals with ASD than those without, reflecting reduced adaptation. Furthermore, individual differences in sustained response in auditory cortex were correlated with severity of ASD symptoms. In accordance with these findings, ASD appears to be associated with increased neural responsiveness. However, this difference, only manifested when repeated stimulation occurred, was confined to specific cortical regions and was specific to the temporal pattern of stimulation. Finally, Jassim et al. [66] conducted a series of robust meta-analyses across 83 experiments from 52 fMRI studies investigating differences between ADS (*n* = 891) and TD (*n* = 967). They found that TD, compared to people with ASD, show greater activity in the prefrontal cortex during perception tasks. More refined analyses revealed that, when compared to typical controls, people with ASD show greater recruitment of the extrastriate V2 cortex during visual processing.

For its high spatial resolution, accessibility, and non-invasive nature, fMRI is frequently used in experimental trials. However, since BOLD contrast is caused by a sluggish response to metabolic changes, its biggest weakness is its low temporal resolution [67]. A brief neural stimulus generates a BOLD response that typically lasts 3 s and peaks 5–6 s after the event. Due to the slowness of these processes, temporal information is heavily blurred. Moreover, since high magnetic fields require customized stimulus delivery and subject-response systems [68], they limit experiment flexibility and complicate multimodal studies that are fundamental to research on SPDs. Lastly, this technique can be difficult to apply to children due to its sensitivity to movement.

## 6. Magnetoencephalography (MEG) Measures

Studies on SPDs that have used magnetoencephalography (MEG) are considerably fewer than those using other techniques. Since MEG has certain complications, such as its extreme sensitivity to movement or its cost, it is not widely used in developmental research. However, it proved to be essential for studying differences in auditory, somatosensory, and visual processing between SPD and NT [69] since it allows to record brain magnetic activity and in identifying patterns of neural oscillations in several bands with highly accurate temporal resolution compared to other neuroimage techniques, such as fMRI. Furthermore, MEG signals are unaffected by distortion [70] and reflect integrated synaptic activity with high fidelity, thus offering a highly accurate measure of brain activity [71]. Kikuchi et al. [72] provided credible evidence indicating that there is reduced connectivity between the left-anterior and right-posterior brain areas in children with ASD (3–7 years old) with SPD. Meanwhile, Dockstader et al. [73] found altered patterns of synchronization and desynchronization in primary and secondary somatosensory cortices in children with ADHD and SPD. Compared to TD, ADHD showed lower oscillatory activity, suggesting a deficit in the perception-to-action system. Marco et al. [74] investigated the neurophysiologic correlates of tactile processing differences in high-functioning children with ASD. They administered a tactile oddball paradigm that presented a slower and a faster condition. Children with ASD showed reduced somatosensory evoked field (SEF) amplitudes as early as 40 ms. This finding was evident in the slow condition but not the fast stimulus presentation, and thus, they proposed a rate-dependent neural mechanism. This difference in early sensory processing may contribute to disrupted higher order processing as well as impact the way an individual approaches their environment. Matsuzaki et al. [75] by means of MEG measured the cortical activation elicited by visual (unisensory) and audiovisual (multisensory) stimuli in children with ASD and TD (7–14 years). The group with ASD demonstrated an increase in the cortical activation in the bilateral insula in response to unisensory stimulus and in the left occipital, right superior temporal sulcus (rSTS), and temporal regions to multisensory movies. The severity of the sensory impairment was correlated with the increased responses. Finally, Demopoulos et al. [76] provided MEG imaging-derived indices of auditory and somatosensory cortical processing in children aged 8–12 years with ASD, those with SPD who do not meet ASD criteria, and typically developing control participants. In all three groups, the magnitude of auditory and tactile responses was comparable, but the M200 latency response from the left auditory cortex was significantly delayed in group with ASD when compared to the TD and SPDs, while the somatosensory response was only delayed when compared to the TD. Participants with SPD showed no significant differences in somatosensory latency between either group, suggesting that they have a phenotype somewhere in between ASD and TD. The specificity of these auditory delays to the group with ASD in addition to their correlation with verbal abilities proves that auditory sensory dysfunction may be implicated in communication symptoms in ASD, motivating further research aimed at understanding the impact of sensory dysfunction on the developing brain.

MEG offers several benefits, including improved safety, fewer limitations, and less signal noise. Moreover, the exquisite temporal and spatial precision of MEG allows us to pinpoint incoming, early sensory activity in the cortex [77]. There is one significant drawback to MEG, however: its sensitivity to head movements, which may result in inaccurate localization of brain activity. For studies with children, this is especially problematic, as they tend to tire more easily under long recording sessions, which also results in less focused attention, more eye movements, and more blinks [71].

## 7. Discussion

The present review highlights that sensory processing is an integrated phenomenon in which stimulus detection is as crucial to its processing as its subsequent elaboration. For this reason, it is important to outline new diagnosis and intervention protocols that consider the influence of environmental factors as well as neural networks involved in sensory processing. Another crucial discussion point refers to the key role that sensory processing represents in neurodevelopment. Researchers agree that difficulties in processing sensory stimuli could represent risk factors in children’s physiological and psychological well-being. Most children who experience SPDs also have other neurodevelopmental disorders, such as ASD and ADHD. Failure in SPDs diagnosis can worsen already-existing symptoms and cause resistance to psychological treatment. SPDs also occurs relatively frequently in children who are considered to be typically developed and do not meet the diagnostic criteria for neurodevelopmental disorders. SPDs have a major impact on the communication and social skills the child needs to communicate with the environment as well as the development of many external and internal disorders [4].

Unfortunately, SPDs are severely under-diagnosed and under-treated due to a lack of precise diagnostic criteria and tools to assess them. Moreover, SPDs patterns may persist into adult life and could influence social functioning and affect individuals’ well-being in adulthood. Therefore, improving diagnostic and therapeutic tools remains a research priority.

The aim of this review was to find a scientific strategy addressing the growing demand for more reliable and valid diagnostic measures for SPDs. These innovations could help researchers and clinicians to distinguish typical and atypical functioning in sensory processes and to identify early phenotypic markers of SPD associated with ASD or other neuropsychiatric conditions even in later adult life. To do so, we reviewed a significant number of studies with different methodologies. Here, we will discuss all the resulting evidence by explaining each method’s weakness and strength.

Many studies have studied SPDs through direct clinical observation and caregiver questionnaires as techniques [23]. Caregiver-report assessment is the most used method to diagnose SPDs. A substantial number of questionnaires have been created in recent decades. Some consider the internal pattern of sensory processing; others consider only a specific behavioural pattern associated with SPD; others focus on the context in which dysfunctional behaviour occurs. All these assessment tools, although varied, have significant methodological limitations. The subjective nature of parent-report data is a major limitation since parents may underestimate sensory processing difficulties, especially for children who are less verbally competent, and they also might not be able to grasp some behavioural responses that are not so striking. Clinical observation of children’s behaviour provides a solution to the data subjectivity problem. Even though there are some clinical observation protocols where children interact with different sensory stimuli, they do not consider the specific contexts in which the interaction takes place. As such, these protocols have low ecological validity, but they omit to consider the heterogeneity of the SPDs phenomenon.

As we have discussed, based on their sensorial threshold and their behavioural response, children with SPDs can be classified into three categories: over-responsive to sensorial stimuli, under-responsive to sensorial stimuli, and sensory craver. Considering the differences between each of these sensory-behavioural patterns, assessment and treatment should also differ from one another. To identify a “specific cognitive-perceptual profile” for each of the three modes of SPDs, it may be important to examine the impact of the environment on the expression of the perceptual deficit. The clinical setting and its lack of ecology might represent a major bias to understand the role of context in the response.

There has been no consideration of neurophysiological patterns associated with SPD in the approaches discussed above. Although psychophysical approaches such as detection threshold measurement may point to potential neural mechanisms for sensory processing deficits in typical and atypical children, they may not effectively separate peripheral from cortical mechanisms or give much information about neurophysiological mechanisms or neuroanatomical loci of differences. The use of ERP evoked potentials allows us to analyse behaviour while recording the brain activity elicited by the stimulus. Thanks to this technique, many studies have shown that in children with SPD (most of whom also had ASD), there is a deficit of cortical inhibitory mechanisms, whose action is essential for the correct processing of sensory stimuli [45]. Moreover, through ERPs it was possible to investigate specific neurophysiological pattern for each sensorial domain [78]. Both DTI and MRS have provided consistent results on alteration in brain connectivity and GABA transmission in children with SPD [51,56]. Similarly, functional magnetic resonance studies (fMRI) have provided great knowledge on SPD neurophysiological mechanism, highlighting sensory cortical inhibitory deficits and abnormal top-down modulation of posterior sensory cortices from prefrontal cortical areas [66].

However, the use of electrophysiological recording and neuroimaging techniques has limitation. A weakness of electrophysiology is its poor spatial resolution, whereas a weakness of neuroimaging is its poor temporal resolution. In detail, EEG cannot pinpoint the exact source of activity and does not distinguish between activities originating in different but adjacent brain areas [79]. In contrast, fMRI can only achieve a limited degree of temporal resolution because intrinsic hemodynamic responses are blurred, and the signal-to-noise ratio is finite [80]. Combining both techniques should be a valuable way of overcoming their limitations. In addition, MEG could also be a valid alternative. Studies with MEG have in fact succeeded in identifying a specific neurophysiological pattern associated with SPDs, characterized by disrupted cortical activity for higher-order processing in each sensory domain. An MEG-related problem could be caused by the difficulty of children with SPDs not moving during MEG analysis. In fact, this technique is very sensitive to cardiac movements and neuromuscular contractions [81]. Nevertheless, MEG is one of the most promising techniques and, used in combination with other instruments, can provide relevant results.

## 8. Future Directions

Finding behavioural and neurophysiological patterns associated with SPD various configurations remains a research priority to this day. Sensory processing is a complex process that involves many levels and mechanisms: perception of stimulus (sensation); processing of stimulus (perception); and integration of perception with emotional, motivational, cognitive, and action processes. A multidimensional approach to studying SPD is necessary due to its complexity. Certainly, the methods we discussed so far are indispensable, but since they each have their disadvantages, it is crucial to understand how to overcome those limitations, including the use of new technologies and innovations. We have seen how SPD is the result of an integrated process involving different cerebral networks and cognitive domains. Hence, it is fundamental to design cognitive, perceptual, and ecological behavioural tasks that also evaluate decision-making and attention processes as well as the ability to mentally represent objects. Furthermore, since we have seen how context plays a key role in the development and maintenance of the perceptual deficits, tasks that measure spatial representation skills should be developed and administered to children with SPDs. Testing for multiple cognitive domains with attentional tests such as the Attentional Network Task (ANT) [82] or the Navon Task [83] as well as mental imagery abilities such as the Mental Imagery Test (MIT) [84] may facilitate the understanding of the processes involved in SPD. Moreover, neuromodulation and virtual reality (VR) techniques are worthy of consideration.

Neuromodulation techniques work by inhibiting, stimulating, regulating, or modulating activity in the nervous system. Brain neuromodulation through cortical and subcortical stimulation has been growing rapidly, with multiple applications in numerous disorders. Its non-invasive nature and reversibility are two of the main reasons for its popularity. As sensory processing requires both stimulus detection and cognitive elaboration, a possible treatment for SPDs should aim at enhancing the child’s cognitive resources. Of all the brain neuromodulation techniques, transcranial magnetic stimulation (TMS) and direct current stimulation (tDCS) are promising, emerging tools for the study and potential treatment of SPDs since their use can modulate the activity of brain networks involved in top-down regulatory mechanisms and measure the information flow between brain posterior regions that serve sensory processes and prefrontal regions that serve executive functions [85,86,87,88]. In general, modifications of brain activity may be sufficient to assist the brain to relearn by inhibiting competing regions, facilitating local activity, or suppressing activity to promote changes. A wide number of paediatric neuropathologies have already been tested with TMS, including perinatal stroke, depression, Tourette syndrome, and ASD. In some cases, paediatric neurologic or neuropsychiatric conditions, such as ASD, ADHD, epilepsy, and cerebral palsy, have also been successfully treated with tDCS [89]. Various single-pulse TMS (sTMS) paradigms have been developed to examine the neurophysiology of ASD, particularly among individuals without intellectual disabilities, to investigate excitability, inhibitory control, and plasticity, respectively. Meanwhile, the use of repetitive TMS (rTMS) as a therapeutic tool is more problematic since few published studies have followed strict clinical trial protocols (e.g., randomization, identification of clear and objective primary endpoints, double-blinding with appropriate sham conditions, sufficient power, etc.); thus, the generalizability to clinical settings is unclear [90]. As for the tDCS, studies on children with ADHD have shown that anodal stimulation in prefrontal cortex does indeed have an improving effect on inhibitory control [91]. Recently, a ground-breaking study by Lazzaro et al. [92] evaluated the effectiveness of various tDCS treatments for children and adolescents with developmental dyslexia, finding consistent improvement in reading skills after treatment. The combination of brain neuromodulation with neuroimaging and/or electrophysiology allows measuring the intermediate neurophysiological effects or causal outcome [93]. Through the use of TMS or tDCS, we will be able to modulate the flow of information between posterior brain regions involved in sensory processing and prefrontal regions involved in executive functioning. Behavioural and electrophysiological changes after neuromodulation would allow us to understand the causal role of altered functional connectivity patterns in children with SPDs.

Virtual reality (VR) emerges as another promising technology to study and treat neurodevelopmental disorders, especially SPDs. VR refers to an advanced, human–computer interface that allows participants to view and move around a virtual environment. They can look at it from different angles, reach into it, grab it, and reshape it. VR capacity to create different scenarios in which the body, environment, and brain are closely related and could improve diagnosis and potential treatment of SPD, especially if integrated with psychophysical paradigms and techniques for exploring physiological and neurological reactivity. Using cognitive and social skill training, rehabilitation through VR has already helped patients with mild cognitive impairment (MCI), schizophrenia, and depression to improve their quality of life. Moreover, it is worth mentioning that VR systems relatively easily deliver virtual environments with well-controlled sensory stimuli [94]. By creating a safe and comfortable virtual environment rich in sensory stimuli, treatment of SPDs can be more effective. In VR, the intensity of sensory stimulation can be gradually increased, sensory input of different types can be included, and these inputs can be administered in different contexts. Thus, through the child’s behavioural and neurophysiological feedback, it will be possible to gradually expose him to more intensive stimulation until he develops an appropriate behavioural response to stimulation. Despite the recent successes of applying VR to treat behavioural disorders [95], there is a lack of studies on SPD. Only Mesa-Gresa et al. [96] highlighted the potential of VR as a therapeutic tool in children with ASD, showing how several clinical trials have recorded improvements in communication, especially social and emotional skills. More studies should focus on the use of VR as a therapeutic tool for children with neurodevelopmental disorders and SPDs.

In conclusion, the need to identify instruments that are more sensitive to the detection of SPD and its symptoms remains a scientific and research priority. The use and application of combined techniques could help to find the neurophysiological correlates associated with the specific behavioural patterns that characterize SPDs. In addition, the use of innovative approaches such as VR and neuromodulation could be promising for treatment perspectives. Lastly, more studies on children with SPD who do not present with ASD or other neurodevelopment disorders are needed.

## Figures and Tables

**Table 1 brainsci-12-01478-t001:** Sum of the principal caregiver-/self-report questionnaires used to assess SPD.

Self-Report Measures	Reference	Type	Items	Age Target	Constructs Assessed
**Sensory Profile** **(SP)**	[26]	Caregiver- report	125 items	3–15 years	Sensory processing, sensory modulation, and behaviour and emotion responses
**The Child Sensory Profile 2** **(SP-2)**	[27]	Caregiver- report	86 items	3–15 years	Sensory systems and behavioural responses
**Short Sensory** **Profile** **(SSP)**	[10,28]	Caregiver- report	38 items	3–15 years	Tactile, taste/smell, and visual/auditory sensitivity
**Sensory Experiences Questionnaire** **(SEQ 3.0)**	[29]	Caregiver- report	105 items	5 months 6 years	Sensory response patterns, enhanced perception, and social or not-social contexts
**The Sensory Rating Scale** **(SRS)**	[30]	Caregiver- report	88–136 items	0–3 years	Sensory responsiveness
**SensOR Inventory** **(SensOR)**	[31]	Caregiver/ self-report	76 items	3–55 years	Sensorial domains, food textures, and eating
**Sensory Processing Measure** **(SPM)**	[32]	Caregiver- report	62 items	3–10 years	Social participation, sensory processing, body awareness, balance, motion, and motor planning
**Infant/Toddler** **Sensory Profile (ITSP)**	[33]	Caregiver- report	36–48 items	0–36 months	Neurological threshold and self-regulation strategies
**Diagnostic Interview for Social and Communication Disorders** **(DISCO)**	[20]	Semi- structured interview with the caregiver	21 items	all ages	Sensory abnormality for different sensorial domains
**Adolescent/Adult Sensory Profile** **(AASP)**	[34]	Self-report	60 items	11–65 years	Sensory responsiveness

## Data Availability

Data are available under request to the corresponding author Laura Mandolesi (laura.mandolesi@unina.it).

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
