# Peer review of "Sensory Processing Disorders in Children and Adolescents: Taking Stock of Assessment and Novel Therapeutic Tools"

_brainsci, 2022, doi:10.3390/brainsci12111478_

Round 1

Reviewer 1 Report

This review mainly discusses the assessment and therapeutic strategies of sensory processing disorders (SPDs) in children and adolescents. The assessment of three SPDs, sensory over-responsiveness, sensory under-responsiveness, and craving for sensory input, are reviewed in detail, and summarized in Table 1. The advantages and limitations of different assessment are comprehensively discussed. Overall, this review provides detailed introduction of diagnostic and research tools and methods for assessing sensory abnormalities.

Comments:

1) The authors may consider revise the title since this paper mainly discusses the sensory processing disorders in children and adolescents.

2) Most of the paragraphs discuss the assessment. The therapeutic strategies are only introduced in the Future Directions of this paper. The authors may consider reorganizing the sections. For example, 3 ‘Caregiver and Self-Report Questionnaires’ to 6 ‘Magnetoencephalography (MEG) measures.’ are assessment methods, which should be include in one section. Then in another section, the author can review the therapeutic strategies, including neuromodulation and virtual reality, respectively.

3) Line 54, use Tourette's Disorder (TD) to help readers from other fields to understand.

4) Please add a title row for Table 1.

5) In table 1, it would be great to include one more column to list the limitations for each assessment approach.

Author Response

The authors may consider revise the title since this paper mainly discusses the sensory processing disorders in children and adolescents.
We thank Reviewer 1 for this suggestion, in the revised version we changed the title in “Sensory processing disorders in children and adolescents:  taking stock of assessment and novel therapeutic tools”.

Most of the paragraphs discuss the assessment. The therapeutic strategies are only introduced in the Future Directions of this paper. The authors may consider reorganizing the sections. For example, 3 ‘Caregiver and Self-Report Questionnaires’ to 6 ‘Magnetoencephalography (MEG) measures.’ are assessment methods, which should be include in one section. Then in another section, the author can review the therapeutic strategies, including neuromodulation and virtual reality, respectively.
While we appreciate the suggestion of Reviewer 1, our goal is to discuss and review the methods and measures available for evaluating SPDs. Virtual reality and neuromodulation, potentially providing support to therapeutic tools already in use to treat the disorder, are suggested, but they are not the main focus of this review. Therefore, we prefer not to change the review's structure, and leave discussion of possible innovative therapeutic techniques to the section 'Future Directions'.

Line 54, use Tourette's Disorder (TD) to help readers from other fields to understand.
In line 54, the abbreviation “TD” refers to children with typical development. In the revised manuscript we have added 'typical development' in full to avoid confusion.

Please add a title row for Table 1.
In the revised manuscript we added a title row for Table 1

In table 1, it would be great to include one more column to list the limitations for each assessment approach
Although we appreciate Reviewer 1's suggestion, Table 1 lists all self-report instruments used for SPDs assessment. The section preceding the table discusses in detail the limitations of this type of instrument. We tried to insert the limits in the table, but we realized that the schematic description could leave critical issues because it is not exhaustive.

Reviewer 2 Report

The current study gives an interesting review on SPDs diagnosis. Nonetheless, there is some visual appeal lacking. I suggest you add some schematics, for example in the introduction section summarizing the different sensory processing disorders, other disorders that they usually appear linked to, their pathophysiology and symptomatology. Also at least in some sections, such as in the case of neuroimaging approaches, you should include some illustrative figures of the data (for example brain scans), that you can get from the cited articles (used with permission, of course). This could improve the readability of your article a lot. Additionally, in section 3 the subsections should be numbered “3.1”, “3.2”, and so on. The same with section 5. Finally, Table 1 lacks a first line where the content of each column is described. After these minor revisions, I advise for the publication of this manuscript.

Author Response

The current study gives an interesting review on SPDs diagnosis. Nonetheless, there is some visual appeal lacking. I suggest you add some schematics, for example in the introduction section summarizing the different sensory processing disorders, other disorders that they usually appear linked to, their pathophysiology and symptomatology.
We appreciate Reviewer 2's interest in our work. There is no doubt that this manuscript is information-intensive, so it may not be visually appealing. Our review was not designed to include schematics on the clinical, symptomatological and pathophysiological features of SPDs as our goal was to provide a critical analysis of assessment tools and to offer innovative solutions that take scientific progress into account when assessing and treating neurodevelopmental disorders. We did not want to add any information that might be too schematic or inaccurate since the focus is not on the clinical features of the disorder.

Also at least in some sections, such as in the case of neuroimaging approaches, you should include some illustrative figures of the data (for example brain scans), that you can get from the cited articles (used with permission, of course). This could improve the readability of your article a lot.
We thank Reviewer 2 for this suggestion, which would certainly make our work much more visually appealing. Unfortunately, this request is time-consuming and would make it difficult to publish the revised manuscript on schedule.

Additionally, in section 3 the subsections should be numbered “3.1”, “3.2”, and so on. The same with section 5.
As suggested, we added subsections in paragraph 3 and 5

Finally, Table 1 lacks a first line where the content of each column is described. After these minor revisions, I advise for the publication of this manuscript.
In the revised manuscript we added a title row for Table 1

Reviewer 3 Report

Dear Authors,

Thank you for the opportunity to revise your work. 

The presented study does not adhere to scientific standards.  Most importantly, it is unclear what this is: certainly not a systematic review, but also not a scoping review. A clear objective and specific aims are missing. 'To conduct a review in order to highlight something' is not an objective of a scientific study

Is this a narrative or a systematic review (SR)? If this was intent to be a systematic review, please revise it accordingly using the PRISMA guideline. 

In methods the Authors reported ”due to the breadth of the topics covered, it is very difficult to set up a systematic review. In this context, it is urgent to first understand the different facets of the disorder and how the different therapeutic techniques can be integrated with each other” . This is not a strong rationale for the science development.  Note that, even if the search string may be complicated by the theme, all the other features of an SR should be considered. 

Here some example:

-       Risk of bias of the self-reported outcome studies should be done. Consider to use the COSMIN,

-       Search terms should be reported, the wording “and so on”, it is not appropriate at all and not worth of a peer-reviewed journal

-       No psychometric proprieties of the self-reported outcome studies has been reviewed or analyzed. 

Authors should have focused on one self-reported outcome or brain imagine

Author Response

The presented study does not adhere to scientific standards.  Most importantly, it is unclear what this is: certainly not a systematic review, but also not a scoping review. A clear objective and specific aims are missing. 'To conduct a review in order to highlight something' is not an objective of a scientific study

Is this a narrative or a systematic review (SR)? If this was intent to be a systematic review, please revise it accordingly using the PRISMA guideline.
In methods the Authors reported ”due to the breadth of the topics covered, it is very difficult to set up a systematic review. In this context, it is urgent to first understand the different facets of the disorder and how the different therapeutic techniques can be integrated with each other” . This is not a strong rationale for the science development.  Note that, even if the search string may be complicated by the theme, all the other features of an SR should be considered.

Here some example:

-       Risk of bias of the self-reported outcome studies should be done. Consider to use the COSMIN,

-       Search terms should be reported, the wording “and so on”, it is not appropriate at all and not worth of a peer-reviewed journal

-       No psychometric proprieties of the self-reported outcome studies has been reviewed or analyzed.

Authors should have focused on one self-reported outcome or brain imagine

We thank Reviewer 3 for his/her suggestions and precious ideas for future development.
It should be noted, however, that ours is a narrative review, as we further specified in the revised version of the manuscript. Due to this, many of the issues raised by Reviewer 3 are not relevant to this type of work.

The purpose of our narrative review is to discuss the different methodologies used to study SPDs. Numerous experimental techniques and paradigms have been used over the years, resulting in often contradictory results. Considering recent clinical evidence linking SPDs to other neurodevelopmental disorders, identifying methods for assessing and discerning the different sensory and behavioral patterns that characterize these deficits remains a scientific priority.

It has been clarified, in the revised version of the manuscript, that our review is not a systematic review, nor does it aim to examine the methods used in SPD research in a systematic way. Rather, it aims to bring together the major scientific evidence on the subject and to point out the strengths and weaknesses of the various perspectives.

It might be interesting to develop a different type of article by focusing on one of the specific aspects outlined by reviewer 3, and we will consider doing so in the future.

To accommodate Reviewer 3's comments, we modified the methods and gave specific guidance on how we conducted the literature review.

Round 2

Reviewer 3 Report

Dear Authors, 

Thank you for the opportunity to revise the manuscript. I feel the work improved. Now, the type of research design, methods and objectives are well defined. 

I do not have any additional comment.

All the best.